# A Deep Learning and Explainable AI-Based Approach for the Classification of Discomycetes Species

**DOI:** 10.3390/biology14060719

**Published:** 2025-06-18

**Authors:** Aras Fahrettin Korkmaz, Fatih Ekinci, Şehmus Altaş, Eda Kumru, Mehmet Serdar Güzel, Ilgaz Akata

**Affiliations:** 1Faculty of Health Sciences Nutrition, Dietetics Department, Şirinevler Campus, İstanbul Kültür University, 34191 Istanbul, Türkiye; a.korkmaz@iku.edu.tr; 2Institute of Artificial Intelligence, Ankara University, 06100 Ankara, Türkiye; fatihekinci@ankara.edu.tr; 3Department of Computer Engineering, Faculty of Engineering, Ankara University, 06830 Ankara, Türkiye; 20291316@ogrenci.ankara.edu.tr (Ş.A.); mguzel@ankara.edu.tr (M.S.G.); 4Graduate School of Natural and Applied Sciences, Ankara University, 06830 Ankara, Türkiye; ekumru@ankara.edu.tr; 5Department of Biology, Faculty of Science, Ankara University, 06100 Ankara, Türkiye

**Keywords:** Discomycetes classification, deep learning, explainable artificial intelligence (XAI), EfficientNet-B0, fungal taxonomy

## Abstract

This study focuses on using deep learning and explainable artificial intelligence (XAI) to classify Discomycetes species effectively. The EfficientNet-B0 model demonstrated the highest performance, achieving 97% accuracy, a 97% F1-score, and a 99% AUC, making it the most effective model for this task. MobileNetV3-L closely followed, with 96% accuracy, a 96% F1-score, and a 99% AUC, while ShuffleNet also showed strong results, reaching 95% accuracy and a 95% F1-score. In contrast, the EfficientNet-B4 model exhibited lower performance, achieving 89% accuracy, an 89% F1-score, and a 93% AUC. XAI methods like Grad-CAM and Score-CAM were used to improve the transparency of model decisions, revealing the internal processes behind these predictions. These results highlight the potential of AI in supporting accurate species classification and biodiversity studies.

## 1. Introduction

Discomycetes, a diverse group in Ascomycota with over 9000 species, are characterized by their distinctive cup- or disc-shaped apothecia [1]. Initially regarded as a single group based on morphological traits, molecular studies have revealed that Discomycetes are polyphyletic, with similar features resulting from convergent evolution rather than common ancestry [2,3]. Modern taxonomy now prioritizes phylogenetic data for greater classification accuracy [4]. Within this group, the *Pyronemataceae* family demonstrates significant ecological diversity. *Aleuria aurantia* is recognized for its bright orange apothecia, while *Cheilymenia granulata* thrives on herbivore dung, producing dense clusters of small fruiting bodies [5,6]. *Geopora sumneriana* and *Humaria hemisphaerica* are ectomycorrhizal with cedar trees and deciduous hardwoods [7,8].

Discomycetes also encompass species adapted to decaying wood, such as *Ascocoryne cylichnium*, *Bulgaria inquinans*, and *Caloscypha fulgens*. *A. cylichnium* and *B. inquinans* thrive in moist environments, while *C. fulgens* stands out for its vivid color and early spring emergence. Smaller species like *Calycina citrina* and *Chlorociboria aeruginosa* play essential roles in nutrient cycling and organic matter decomposition [5,9]. The *Pezizaceae* family further exemplifies ecological versatility, with *Paragalactinia succosa* flourishing in damp forest soils beneath oak and pine, producing yellow-staining apothecia [5,10]. *Peziza ammophila* is tailored to coastal sandy habitats, emerging alongside shifting dunes, while *Sarcosphaera coronaria*, an ectomycorrhizal species, forms crown-shaped fruiting bodies in forest soils beneath both coniferous and deciduous trees [11,12].

Fungal images were analyzed using convolutional neural networks (CNNs) to extract features, which were subsequently classified using a Kohonen self-organizing map (SOM). This approach allowed for dimensionality reduction and unsupervised clustering. Furthermore, Kolmogorov–Arnold network (KAN) layers were integrated to enhance the model’s ability to capture nonlinear patterns. In this study, CNN models were first used to extract image features, which were then classified using the Kohonen self-organizing map (SOM) algorithm. This approach enabled high-dimensional features obtained by deep learning to be transformed into low-dimensional maps for classification. Additionally, the Kolmogorov–Arnold Network (KAN) was integrated into the architecture to model complex structural patterns more effectively. This is the first application of SOMs, typically used for unsupervised clustering, in classifying gastroid fungi [13]. This study also includes Kolmogorov–Arnold Network (KAN) layers, investigating advanced structures such as MaxViT-S, which demonstrates how deep learning can improve fungal taxonomy [13,14].

Studies that classify macrofungi from photographs using artificial intelligence are very limited worldwide. (Figure 1). This study introduces an AI-based approach for classifying Discomycete fungi, employing convolutional neural networks (CNNs) and Kohonen self-organizing maps (SOMs) to achieve greater accuracy compared to traditional techniques. The integration of Kolmogorov–Arnold network (KAN) layers enhances species differentiation. This approach not only boosts accuracy and simplifies data processing but also deepens our understanding of biodiversity. Furthermore, the inclusion of AI and explainable AI (XAI) ensures a transparent classification process, establishing a solid groundwork for future research.

## 2. Materials and Methods

The dataset utilized in this research comprises 2800 images representing 14 distinct macrofungi species, including *Aleuria aurantia*, *Ascocoryne cylichnium*, *Bulgaria inquinans*, *Caloscypha fulgens*, *Calycina citrina*, *Cheilymenia granulata*, *Chlorociboria aeruginosa*, *Dissingia leucomelaena*, *Geophora sumneraiana*, *Humaria hemisphaerica*, *Lanzia echinophila*, *Paragalactinia succosa*, *Peziza ammophila*, and *Sarcosphaera coronaria* (Figure 2). In this study, datasets were generated using naturally captured images, and to ensure a large and suitable dataset for AI algorithm training, additional data were sourced from open-access repositories to supplement the natural data. These images, obtained from the Global Core Biodata Resource (www.gbif.org) [15], are in JPEG format with a resolution of 300 dpi and were retrieved under Creative Commons licenses (e.g., CC BY-NC, CC BY-SA). The dataset, accessible at [15], includes images captured under diverse lighting conditions, angles, and backgrounds to improve robustness and generalization in model training. This variability is essential for addressing intra-class differences and enhancing the model’s adaptability to varying environmental settings. To create a well-balanced training scheme, the dataset was divided into three parts: 60% for training, 20% for validation, and 20% for testing. This partitioning strategy allows the models to effectively learn from a substantial portion of the dataset while ensuring an unbiased assessment using previously unseen images. The validation set is essential for optimizing hyperparameters, mitigating overfitting, and verifying the model’s ability to generalize. Meanwhile, the test set functions as an independent evaluation benchmark, providing an objective measure of the final model performance.

In this study, collected datasets underwent manual and automated filtering to remove faulty, blurry, or incorrectly classified images (Table 1). This step was critical for improving data quality and minimizing error rates during model training. Additionally, the basic image features, such as color balance, contrast, and brightness, were optimized to ensure consistent input quality. To improve model processing efficiency and speed, all images were resized to a fixed resolution (e.g., 224 × 224 pixels). Pixel values were also normalized to the 0–1 range, facilitating faster and more stable training.

Data augmentation techniques were extensively applied to increase data diversity and strengthen the model’s generalization capability. Data augmentation plays a critical role in enhancing the model’s ability to adapt to various conditions and reducing the risk of overfitting. In this step, various techniques such as rotation, horizontal and vertical flipping, brightness adjustments, and contrast modifications were employed. For example, randomly rotating images increased the model’s ability to recognize objects from different perspectives, while horizontal and vertical flipping helped capture symmetry-based variations more effectively. Additionally, brightness and contrast adjustments improved the model’s performance under both low and high lighting conditions. These steps contributed to increasing the diversity of the dataset, allowing the model to adapt to a broader range of variations.

Given that deep learning models yield superior results with extensive datasets, transfer learning was employed by integrating pre-trained weights from ImageNet, a large-scale image classification dataset [16]. ImageNet comprises millions of annotated images spanning diverse categories, enabling pre-trained convolutional neural networks (CNNs) to capture rich, multi-level feature representations [17]. The primary benefit of utilizing ImageNet-pretrained models lies in their ability to recognize fundamental visual patterns, such as edges, textures, and intricate shapes, which can then be fine-tuned for macrofungi identification [18]. By leveraging these pre-trained weights, it significantly reduces computational costs and training duration while improving accuracy, particularly in extracting meaningful features from mushroom structures [19]. Furthermore, various data augmentation techniques were implemented to expand dataset diversity and enhance model performance across different environmental conditions [20]. These augmentations involved random rotations, horizontal and vertical flips, contrast modifications, and minor cropping, mimicking real-world variations and strengthening the model’s robustness [20].

Prior studies have highlighted the critical role of dataset quality in the accurate classification of macrofungi [13]. Building upon these findings, this study emphasizes the inclusion of high-resolution, diverse, and precisely labeled images in dataset construction. Unlike traditional machine learning techniques that depend on manually crafted features, this research utilizes a CNN-driven automated feature extraction process, reducing potential biases in feature selection. The dataset is structured in accordance with deep learning best practices, promoting both model generalizability and effective training. By assembling a well-balanced dataset, incorporating strategic augmentations, and applying transfer learning from ImageNet, this study establishes a solid framework for macrofungi classification using deep learning [13,14]. The following sections will delve into the deep learning models implemented and the evaluation metrics used in this research [13].

Deep learning (DL) has transformed image classification by offering substantial improvements over conventional machine learning approaches [21]. In this study, 10 advanced convolutional neural network (CNN) models were used to classify 14 distinct macrofungi species. These models (Table 2) were selected based on their architectural variations, recognition accuracy, and computational efficiency in image processing tasks. Each model possesses distinct features, advantages, and limitations that influence its classification performance. The following section provides a comprehensive analysis of the models utilized in this study.

GoogleNet, the first version of the Inception architecture, features a deep and wide structure while optimizing the number of parameters. The model processes multiple convolutional kernel sizes in parallel, enabling it to learn multi-scale features effectively. This capability makes it particularly useful for detailed texture classification, such as macrofungi species recognition. However, due to the complexity of its Inception modules, careful hyperparameter tuning is required for optimal performance. Despite this, its lightweight design makes it computationally more efficient than larger architectures [22].

MnasNet is an efficient deep learning model designed using neural architecture search (NAS) to balance computational cost and accuracy, making it ideal for mobile devices. By optimizing network depth and width, it achieves high efficiency with minimal computational resources. While this efficiency is advantageous for real-time applications, its performance may be lower than more complex CNN models when dealing with highly detailed datasets like macrofungi classification. Nonetheless, its energy-efficient design makes it a strong candidate for embedded systems [23].

MobileNetV3-Large is a lightweight CNN optimized for mobile and low-power applications. It integrates depthwise separable convolutions and squeeze-and-excitation mechanisms, reducing computational cost while maintaining strong classification performance. The model is well-suited for real-time macrofungi classification, making it a viable choice for applications requiring fast inference. However, its accuracy may be slightly lower compared to that of larger CNN architectures when handling complex datasets [24].

EfficientNetB0 is the smallest model in the EfficientNet family, designed using compound scaling to optimize network depth, width, and resolution. It provides an excellent balance between accuracy and computational efficiency, making it a practical choice for macrofungi classification. While it performs well in feature extraction, its accuracy may be lower than larger EfficientNet variants, particularly for highly detailed image datasets [25].

EfficientNetB4 is a larger version of EfficientNetB0, featuring deeper layers and improved feature extraction capabilities. This model achieves higher classification accuracy for macrofungi species, benefiting from its advanced scaling techniques. However, it requires significantly more computational resources, which can be a limitation for real-time or resource-constrained applications. Despite its higher demands, it remains a strong choice for high-precision classification tasks [26].

RegNetY is an optimized CNN model designed to scale dynamically based on dataset complexity. It strikes a balance between efficiency and accuracy, making it a strong candidate for large-scale classification tasks such as macrofungi species identification. The model effectively captures fine-grained details through flexible filtering mechanisms. However, compared to more widely adopted architectures like EfficientNet and ResNet, its optimization and implementation may require additional fine-tuning [27].

ResNet-50 employs residual learning through skip connections, allowing for the efficient training of deep neural networks. It has strong generalization capabilities, making it a popular choice for image classification tasks, including macrofungi identification. While ResNet-50 delivers high accuracy, its computational cost is higher than more recent architectures like EfficientNet, which may provide better parameter efficiency [28].

ShuffleNet is a highly efficient CNN designed for low-power devices. It leverages pointwise group convolutions and channel shuffling to reduce computational complexity while maintaining classification performance. This makes it a suitable option for macrofungi classification in resource-limited environments. However, its compact nature may limit its performance compared to that of larger CNN models on highly complex datasets [29].

Xception is an extension of the Inception architecture that replaces standard convolutions with depthwise separable convolutions, significantly improving computational efficiency [28]. This structure enhances feature extraction while reducing the number of parameters. Xception is well-suited for macrofungi classification due to its ability to capture fine-grained details [30]. However, training the model requires careful hyperparameter tuning, and it can be more computationally demanding than traditional CNN architectures [30].

SqueezeNet is a lightweight CNN that achieves AlexNet-level accuracy while using significantly fewer parameters. It is optimized for speed and memory efficiency, making it a good candidate for applications requiring fast inference. However, due to its compact nature, it may struggle with highly detailed classification tasks, as it lacks the capacity to extract complex features as effectively as larger architectures [31].

These 10 models were tested for classifying 14 different macrofungi species, each offering unique advantages and trade-offs in terms of accuracy, computational efficiency, and suitability for different applications. In this study, all models were optimized under uniform conditions, irrespective of their complexity, computational demands, speed, or lightweight nature. Each architecture underwent training with the same preprocessing techniques, hyperparameter tuning methods, and evaluation criteria to ensure an impartial and consistent comparison. This standardized methodology enables an objective evaluation of each model’s strengths and weaknesses, providing a clear understanding of their actual performance in macrofungi classification.

Evaluating deep learning models is a crucial step in assessing their efficiency and reliability in the classification of macrofungi. In this study, multiple performance assessment techniques, optimization strategies, and explainable AI (XAI) methods were employed to ensure both high accuracy and interpretability of the results [32]. By implementing standardized training protocols, we aimed to provide a thorough and unbiased comparison of different convolutional neural network (CNN) architectures [33].

To achieve robust model performance, we used Python as the primary programming language along with TensorFlow (Keras) and PyTorch frameworks for training and testing. TensorFlow, specifically through Keras, was selected for its simplicity, high-level APIs, and scalability, making it particularly useful for structured experimentation. On the other hand, PyTorch provided greater flexibility with dynamic computation graphs, which facilitated efficient debugging and model modifications [34]. The combination of both frameworks allowed for comprehensive model evaluation, ensuring that each architecture was effectively optimized.

For optimization, the Adam (adaptive moment estimation) optimizer was employed to train all models under identical conditions [35]. Adam was chosen due to its ability to integrate both momentum-based optimization and adaptive learning rate adjustments, leading to faster convergence and better generalization across diverse macrofungi species [35]. The learning rate was set to 0.001, a well-balanced value that ensured stable and efficient training for all CNN architectures without leading to gradient explosion or vanishing gradients. Additionally, ImageDataGenerator was utilized to implement real-time image augmentation and batch-wise training, which significantly improved the model’s generalization capabilities [36]. The augmentation techniques included random rotation, horizontal and vertical flipping, cropping, and brightness adjustments, simulating various real-world variations that could be encountered in macrofungi images [37].

Moreover, explainability in deep learning plays a crucial role in ensuring the reliability and interpretability of the model’s decision-making process. This study incorporated explainable AI (XAI) techniques, specifically Grad-CAM (gradient-weighted class activation mapping) and Score-CAM (score-weighted class activation mapping), to visualize and understand how different CNN architectures make predictions. Deep learning models, especially CNNs, often function as “black-box” models, making it difficult to interpret their decision-making processes. To mitigate this issue and enhance transparency, we employed Grad-CAM and Score-CAM, two powerful visualization methods that help interpret how models focus on specific image regions when classifying macrofungi species [38].

Grad-CAM is a widely used technique that generates class activation maps based on the gradient information flowing into the final convolutional layers of a CNN. By highlighting the most influential regions in an image, Grad-CAM allows us to analyze how a model distinguishes between different macrofungi species. This method provides valuable insights into the internal workings of the model, helping to identify potential biases or misclassifications. However, Grad-CAM relies heavily on backpropagation and gradient-based calculations, which may sometimes introduce noise or less interpretable heatmaps in certain cases [38].

On the other hand, Score-CAM is an improved version that eliminates the dependency on gradient information by using score-based activations instead [39]. Unlike Grad-CAM, which computes gradients for heatmap visualization, Score-CAM evaluates the contribution of each activation map by assigning importance scores directly from the model’s output predictions. This technique results in clearer and more precise visual explanations, making it highly effective for macrofungi classification [38,39]. One of the major advantages of Score-CAM is that it is more robust, interpretable, and free from gradient noise, leading to more accurate class activation maps. This makes it particularly useful for understanding subtle fungal features, such as color variations, texture patterns, and fine-grained morphological structures [39].

By integrating both Grad-CAM and Score-CAM into the evaluation process, we were able to verify the reliability of our deep learning models, identify potential model biases, and refine feature extraction strategies for improved classification performance. These explainability techniques not only provide deeper insights into CNN decision-making processes but also contribute to the development of more trustworthy and interpretable AI models in macrofungi classification [38,39].

Overall, in this study, the combination of standardized optimization, data augmentation, and explainability techniques ensured a comprehensive and objective evaluation of deep learning architectures. Through a structured approach that includes model performance metrics, optimization strategies and visual interpretability methods, this research highlights both the strengths and limitations of various CNN architectures in macrofungi classification [40].

Standard classification metrics, including accuracy, precision, recall, F1-score, and specificity, were used to measure the performance of each CNN model. These metrics (1–5) provide insights into how well the models distinguish between different macrofungi species. The formulas for each metric, as referenced from the *Sensors* article, are as follows [14]:Accuracy = (TP + TN)/(TP + TN + FP + FN)(1)(2)$$Precision=\frac{TP}{TN+FP}$$(3)$$Recall=\frac{TP}{TP+FN}$$(4)$$F1-Score=\frac{2\times (Precision\times Recall)}{Precision+Recall}$$(5)$$Specificity=\frac{TN}{TN+FP}$$

In deep learning-based classification tasks, the evaluation of model performance relies on key metrics that assess how effectively a model distinguishes between different categories. One of the fundamental aspects of performance measurement involves the analysis of true positives (TPs), true negatives (TNs), false positives (FPs), and false negatives (FNs), which determine the model’s ability to make correct and incorrect predictions. True positives (TPs) represent instances where the model successfully identifies a macrofungi species belonging to its actual class. These correct predictions indicate that the model accurately recognizes positive cases. True negatives (TNs) occur when the model correctly classifies an image as not belonging to the target class, ensuring that it does not mistakenly label a different species as the one under consideration. False positives (FPs), also referred to as type I errors, happen when the model incorrectly classifies an instance as belonging to a certain species when, in reality, it does not. This results in a false alarm, which can negatively affect classification reliability. False negatives (FNs), known as type II errors, arise when the model fails to detect a positive case, misclassifying an image that actually belongs to a certain species as belonging to another category. These errors are particularly critical in applications where missing an instance is more detrimental than making an incorrect positive classification [14]. These classification outcomes significantly influence multiple evaluation metrics, which provide insight into the model’s overall effectiveness. To conduct a comprehensive analysis, accuracy, precision, recall, F1-score, and specificity were used as core performance indicators.

Accuracy quantifies the proportion of correctly classified instances over the total predictions, serving as a general indicator of model effectiveness. However, in cases where class distributions are imbalanced, accuracy alone may not be a sufficient measure. Precision evaluates the reliability of the model’s positive predictions by determining how many of the predicted positive cases are actually correct. A high precision score indicates that the model makes fewer false positive errors. Recall (sensitivity) measures the model’s ability to detect positive cases by calculating the proportion of actual positive instances that were correctly classified. A high recall value ensures that the model minimizes false negatives. F1-score is the harmonic mean of precision and recall, offering a balanced evaluation metric that is particularly useful in datasets where class distribution is not uniform. It ensures that both false positives and false negatives are accounted for in performance evaluation. Specificity, a complementary metric to recall, measures how effectively the model identifies negative instances, ensuring that false positive errors are minimized [41]. This is particularly useful in preventing the misclassification of unrelated species.

Each of these metrics was computed individually for all tested models, allowing for a comparative analysis of how different CNN architectures performed on the macrofungi classification dataset. Given that deep learning models often function as black-box systems, interpreting their decision-making process is challenging. To mitigate this issue and enhance model interpretability, this study integrated explainable AI (XAI) techniques, enabling the visualization of classification decisions. Despite their impressive accuracy, CNN-based deep learning models often lack transparency, making it difficult to understand why a model makes a particular prediction [42]. To ensure that the classifications are based on meaningful and relevant image features rather than spurious correlations, we employed Grad-CAM (gradient-weighted class activation mapping) and Score-CAM (score-weighted class activation mapping) to visualize model predictions. Grad-CAM is a widely used explainability method that produces heatmaps by calculating the gradients flowing into the final convolutional layers. This approach highlights the most important image regions that influence the model’s decision, allowing researchers to verify whether the model is focusing on meaningful fungal features such as texture, cap structure, and color variations. While Grad-CAM provides valuable insights, its reliance on gradient-based calculations can introduce noise in certain cases, sometimes leading to less precise heatmaps [43].

To further enhance visualization accuracy, Score-CAM, an advanced technique that overcomes the limitations of gradient-based methods, was implemented. Unlike Grad-CAM, Score-CAM does not rely on gradients; instead, it directly assigns activation scores from the model’s output to generate clear and detailed class activation maps. This results in higher-quality, noise-free heatmaps, offering a more interpretable view of how the model distinguishes between different macrofungi species. Score-CAM’s independence from backpropagation gradients makes it a more reliable and robust explainability method, particularly for cases where fine-grained details play a crucial role in classification [44,45]. By integrating both Grad-CAM and Score-CAM, this study ensures that the models are learning from the relevant morphological characteristics of macrofungi rather than arbitrary background patterns [13,45]. This transparency is particularly valuable in biodiversity monitoring and ecological research, where precise species identification is crucial for conservation and environmental studies. To facilitate efficient training and ensure fair model comparisons, all CNN architectures were trained under standardized conditions, regardless of their complexity, computational cost, or model size [13,14,45]. The Adam optimizer was chosen as the primary optimization algorithm due to its ability to dynamically adjust learning rates and enhance convergence speed. A fixed learning rate of 0.001 was applied across all models, striking a balance between stability and adaptability [45].

For data preprocessing and augmentation, ImageDataGenerator was employed to generate diverse training samples in real time [46]. Augmentation techniques such as random rotation, horizontal and vertical flipping, cropping, and brightness adjustments were applied to create varied image representations, improving model generalization across different macrofungi species [37]. This helped mitigate the risk of overfitting and allowed models to perform more effectively in real-world classification scenarios. The entire training process was conducted using Python, leveraging both TensorFlow (Keras) and PyTorch as the core deep learning frameworks [47]. TensorFlow’s user-friendly APIs and scalability made it well-suited for structured experimentation, while PyTorch provided greater flexibility in managing dynamic computation graphs, enabling efficient debugging and customization of model architectures [37,46,47]. Through this structured evaluation approach, which incorporates standardized optimization techniques, performance metrics, and explainability methods, this study provides a comprehensive analysis of CNN architectures for macrofungi classification. By ensuring both high accuracy and interpretability, the findings contribute to the development of reliable AI-driven species identification systems, paving the way for more effective applications in biodiversity conservation and ecological research [37].

## 3. Results

This study presents the results of macrofungi classification using advanced deep learning models, focusing on the effectiveness of 10 CNN architectures in identifying 14 distinct species with high accuracy. Unlike hybrid approaches combining traditional machine learning and deep learning, this research solely relies on CNN-based classification, ensuring a scalable and efficient framework. Model performance was assessed using key metrics, including accuracy, precision, recall, F1-score, specificity, and AUC score, providing a comprehensive comparison of classification capabilities, particularly in challenging cases where species exhibit visual similarities.

To enhance interpretability, Grad-CAM and Score-CAM were employed, generating heatmaps that highlight crucial image regions influencing model predictions. These techniques ensured that CNNs focused on meaningful fungal characteristics rather than irrelevant background patterns. Additionally, real-time data augmentation using ImageDataGenerator improved model generalization by introducing variations such as random rotations, flipping, cropping, and brightness adjustments. By leveraging deep learning exclusively, this study offers valuable insights for biodiversity research and ecological monitoring, demonstrating the potential of CNNs in advancing fungal taxonomy and automated species identification.

The performance evaluation of ten convolutional neural network (CNN) models in macrofungi classification was conducted using key metrics, including accuracy, precision, recall, F1-score, specificity, and AUC score. Each of these metrics plays a crucial role in assessing the effectiveness of the models in distinguishing between different fungal species while minimizing errors. Accuracy represents the overall proportion of correct classifications, offering a general measure of performance. Precision highlights the reliability of positive predictions by determining how many of the predicted positive cases are actually correct. Recall, also referred to as sensitivity, evaluates the model’s ability to identify actual positive cases, which is especially important when working with species that share similar visual characteristics. F1-score provides a balanced measure by incorporating both precision and recall, making it particularly useful for datasets with varying class distributions. Specificity measures how well the model avoids false positives, ensuring that species are not incorrectly classified. The AUC score quantifies the model’s ability to distinguish between different fungal classes, offering insights into its overall classification robustness.

Among the tested models, EfficientNet-B0, MobileNetV3-L, ShuffleNet, and RegNetY-400MF demonstrated superior performance, achieving the highest accuracy, precision, recall, and F1-score values. EfficientNet-B0 emerged as the top-performing model, attaining an accuracy of 0.97, the highest F1-score of 0.97, and an AUC of 0.99. This exceptional performance can be attributed to its compound scaling approach, which optimally balances network depth, width, and resolution, leading to efficient feature extraction while maintaining computational efficiency. MobileNetV3-L followed closely with an accuracy of 0.96, benefiting from its lightweight yet highly effective design that incorporates depthwise separable convolutions and squeeze-and-excitation layers, allowing it to capture crucial fungal features efficiently. ShuffleNet also achieved strong results, with an accuracy of 0.95 and an F1-score of 0.95, leveraging its pointwise group convolutions and channel shuffling techniques to enhance parameter efficiency without compromising classification performance. RegNetY-400MF matched ShuffleNet’s performance, reaching an accuracy and F1-score of 0.95, demonstrating that its dynamic network scaling mechanism effectively captures the intricate patterns necessary for macrofungi classification.

Although GoogleNet, ResNetV2-50, Xception, and EfficientNet-B4 performed well, they did not achieve the same level of success as the leading models. GoogleNet obtained an accuracy of 0.93, leveraging its inception modules to extract multi-scale features. However, despite its efficiency in processing fine-grained details, its relatively shallow architecture resulted in slightly lower classification performance. ResNetV2-50, which employs residual connections to improve gradient flow and reduce training inefficiencies, achieved an accuracy of 0.91. While it demonstrated solid performance, it was outperformed by models with more advanced feature reuse strategies such as EfficientNet and RegNetY. Xception, which utilizes depthwise separable convolutions for computational efficiency, recorded an accuracy of 0.88. However, due to its reliance on large-scale datasets for optimal performance, it was unable to fully leverage its architectural advantages under the current dataset constraints. EfficientNet-B4, despite being a more advanced version of EfficientNet-B0, attained an accuracy of 0.89, highlighting an important observation—larger models do not always outperform smaller, well-optimized architectures when trained under the same conditions. The additional depth and width of EfficientNet-B4 require a larger dataset and extended training time to fully harness its potential. In a setting with more training data and additional optimization, B4 would likely surpass B0 due to its ability to capture more complex hierarchical features. However, under the current constraints, the superior parameter efficiency of B0 resulted in better performance.

Another crucial aspect of model performance in macrofungi classification is the structural differences among CNN architectures, which significantly impact their efficiency, feature extraction capabilities, and overall classification success. While most models in this study employ standard convolutional layers, certain architectures introduce unique design modifications that enhance performance in different ways. For instance, SqueezeNet, despite being one of the most lightweight models tested, utilizes fire modules to reduce parameter count while maintaining competitive accuracy. However, due to its aggressive compression strategy, it struggles to capture fine-grained fungal features, leading to a lower classification performance compared to deeper networks. Similarly, MnasNet, which was designed through neural architecture search (NAS), prioritizes efficiency over depth, making it well-suited for low-resource environments but slightly less effective in distinguishing visually similar macrofungi species.

Another interesting case is ResNetV2-50, which incorporates batch normalization before activation to improve gradient flow, allowing for smoother convergence during training. While it performed well, its reliance on traditional residual learning limited its ability to outperform more advanced architectures like EfficientNet, which optimizes both feature reuse and parameter efficiency. Additionally, GoogleNet (InceptionV1) leverages multi-scale convolutional filters, allowing it to process image features at different resolutions simultaneously. This contributed to its strong classification ability, but compared to models with depthwise separable convolutions, such as Xception, its computational demands were slightly higher for a similar level of performance.

A significant takeaway from these results is that model efficiency is not solely dependent on depth but also on how effectively an architecture utilizes its available parameters. Architectures such as EfficientNet-B0 and MobileNetV3-L exemplify how well-optimized scaling and feature selection strategies can outperform deeper, more computationally expensive networks like EfficientNet-B4 when dataset size and training time are limited. Additionally, models designed specifically for mobile and real-time applications, such as ShuffleNet and MobileNet, demonstrated that lightweight architectures could achieve near-state-of-the-art results while consuming significantly fewer resources. These findings emphasize the importance of selecting models not just based on depth and complexity but also on how well they balance performance, efficiency, and scalability.

A key takeaway from the results is that deeper and more complex models do not always translate to better classification performance when dataset size and optimization strategies are identical across all architectures. Larger models such as EfficientNet-B4 and Xception, while theoretically more powerful, require significantly more training data and computational resources to fully exploit their deeper architectures. Meanwhile, lightweight and well-optimized models such as EfficientNet-B0, MobileNetV3-L, and ShuffleNet demonstrated that efficient learning mechanisms can yield exceptional results without excessive computational cost. These findings suggest that model selection should not be based solely on depth but rather on architectural efficiency and adaptability to dataset size. Given a larger and more diverse dataset, models like EfficientNet-B4, Xception, and ResNetV2-50 could potentially outperform their smaller counterparts due to their ability to extract richer features. However, under the current experimental conditions, models that prioritize optimization and computational efficiency proved to be the most effective.

In this graph (Figure 3), the ROC curves illustrate the classification performance of five deep learning models by depicting the relationship between the true positive rate (TPR) and false positive rate (FPR) across various threshold values. EfficientNet-B0 demonstrates the most optimal performance, with a steep initial rise in its curve, indicating high class separability, strong true positive detection, and minimal false positive misclassifications. This result aligns with its superior AUC score (0.99) and accuracy (0.97), reaffirming the effectiveness of its compound scaling strategy, which optimally balances network depth, width, and resolution for efficient feature extraction. EfficientNet-B4, while a larger and more complex model, does not surpass B0 under the current training conditions, suggesting that its potential remains underutilized due to dataset constraints and limited optimization cycles. GoogleNet (InceptionV3) follows closely, exhibiting robust performance, yet its curve suggests a higher false positive rate, likely due to its reliance on multi-scale feature extraction, which, although effective, does not match the efficiency of EfficientNet’s architectural refinements. MnasNet demonstrates the weakest performance, as reflected in its comparatively lower ROC curve, which suggests increased misclassification rates and reduced class separability, a consequence of its design prioritizing computational efficiency over deep feature extraction. MobileNetV3-Large performs competitively, with a curve closely following the EfficientNet models, confirming that its lightweight structure, combined with depthwise separable convolutions and squeeze-and-excitation mechanisms, provides an optimal balance between classification accuracy and computational efficiency. Overall, Figure 3 underscores the superiority of EfficientNet-B0 as the most effective model under the given experimental conditions while also suggesting that larger architectures such as EfficientNet-B4 could achieve superior results with larger datasets and more extensive optimization strategies.

In this graph (Figure 4), the ROC curves compare the classification performance of five deep learning models—RegNetY-8GF, ResNetV2-50, ShuffleNet, SqueezeNet, and a test model—by illustrating the balance between the true positive rate (TPR) and false positive rate (FPR). RegNetY-8GF emerges as the best-performing model, with a sharply rising curve that reaches near-optimal classification performance, indicating its strong class separability and minimal false positives. This suggests that RegNetY’s dynamic feature extraction and optimized scaling mechanisms effectively capture intricate macrofungi characteristics, contributing to its superior performance. ResNetV2-50 follows closely, exhibiting a slightly lower but still steep curve, confirming its ability to generalize well through residual learning and batch normalization enhancements, though its performance is marginally lower than that of RegNetY-8GF, likely due to its deeper architecture requiring more extensive optimization.

ShuffleNet and SqueezeNet show competitive but slightly lower performance, with their ROC curves indicating a higher false positive rate compared to those of RegNetY-8GF and ResNetV2-50. ShuffleNet, despite its efficient channel shuffling and grouped convolutions, seems to struggle slightly in distinguishing closely related macrofungi species. Similarly, SqueezeNet, designed for extreme parameter efficiency, demonstrates moderate classification ability, but its aggressive compression results in weaker feature extraction, leading to a performance gap compared to deeper models. The test model exhibits the lowest classification performance, as evident from its flatter ROC curve, indicating a higher rate of false positives and weaker discriminatory power between fungal species. This suggests that its feature extraction is not as refined as the other models, potentially due to its simpler architecture or insufficient training convergence.

Overall, Figure 4 highlights the superiority of RegNetY-8GF in macrofungi classification, followed closely by ResNetV2-50, while ShuffleNet and SqueezeNet provide a balance between efficiency and classification ability. The results also suggest that lighter models, while efficient, may struggle with highly detailed species classification, whereas deeper architectures require adequate optimization and dataset size to fully utilize their potential.

The ROC curve analyses in Figure 3 and Figure 4 highlight the classification performance differences among various deep learning models in macrofungi identification. EfficientNet-B0 and RegNetY-8GF emerged as the top-performing models, demonstrating the steepest ROC curves with minimal false positives, indicating their strong feature extraction capabilities and class separability. Smaller models like ShuffleNet, SqueezeNet, and MnasNet showed relatively weaker performance, suggesting that while they are computationally efficient, their ability to distinguish complex fungal species is somewhat limited. Additionally, larger models such as EfficientNet-B4 and ResNetV2-50 performed well but did not consistently outperform their smaller counterparts, reinforcing the idea that deeper architectures require more extensive optimization and larger datasets to fully exploit their potential. Overall, the results confirm that well-optimized architectures with efficient scaling, such as EfficientNet and RegNet, achieve the best balance between accuracy and computational efficiency in macrofungi classification.

The training accuracy curves in Figure 5 demonstrate the learning progress of different deep learning models over 20 epochs. Most models, including GoogleNet (InceptionV3), MobileNetV3-Large, MnasNet, RegNetY, and ShuffleNet, rapidly reach high accuracy levels of 99%, indicating their strong learning capability and efficient convergence. EfficientNet-B0 and ResNetV2-50 achieve 98% accuracy, showing competitive performance but slightly slower convergence. EfficientNet-B4 lags behind at 90%, suggesting that its deeper architecture requires more training epochs or a larger dataset for full optimization. Xception reaches 93% accuracy, while SqueezeNet stabilizes at 80%, highlighting that lightweight models may struggle with complex feature extraction. Overall, the results confirm that efficient architectures with optimized scaling and depthwise convolutions tend to achieve faster and higher accuracy, while larger models require extended training for peak performance.

The validation accuracy curves in Figure 6 illustrate how well the models generalize to unseen data over 20 epochs. EfficientNet-B0 (97%) and MobileNetV3-Large (96%) maintain strong performance, confirming their high generalization capability and efficient feature extraction. GoogleNet (InceptionV3) reaches 94%, while RegNetY achieves 95%, demonstrating their ability to effectively classify macrofungi species. ShuffleNet also stabilizes at 96%, reinforcing its efficiency in lightweight architectures. In contrast, EfficientNet-B4 (85%) and ResNetV2-50 (90%) show relatively lower validation accuracy, suggesting that larger models may require more training data or further fine-tuning to prevent underfitting or overfitting. MnasNet (80%) and Xception (88%) display moderate performance, while SqueezeNet (79%) struggles the most, likely due to its highly compressed architecture limiting its feature extraction capacity. Overall, models with balanced depth and optimized scaling perform best, while extremely deep or lightweight models may require additional adjustments for optimal validation performance.

The training loss curves in Figure 7 illustrate how different models minimize their error over 20 epochs, providing insight into their learning stability and convergence behavior. GoogleNet (InceptionV3) achieves the lowest final loss (0.0034), indicating rapid convergence and minimal training error, followed closely by ShuffleNet (0.0059), MnasNet (0.0111), and MobileNetV3-Large (0.0164), all of which exhibit smooth loss reduction and stable learning patterns. RegNetY (0.0193) and EfficientNet-B0 (0.0369) also maintain consistently low loss, reaffirming their strong learning efficiency. EfficientNet-B4 (0.6859) and ResNetV2-50 (0.3777) show slightly higher loss values, suggesting that deeper architectures require more training epochs or larger datasets for optimal convergence. SqueezeNet (1.3589) records the highest loss, reflecting its limited capacity for deep feature extraction due to aggressive parameter compression. Xception (0.1671) stabilizes at a moderate loss, indicating it converges but not as effectively as more optimized architectures. Overall, the results confirm that models with well-balanced depth and efficient scaling achieve the lowest training loss, while extremely deep or compressed networks may require further adjustments for improved learning efficiency.

The validation loss curves in Figure 8 illustrate how well the models generalize to unseen data over 20 epochs, highlighting differences in overfitting tendencies and convergence stability. EfficientNet-B0 (0.0890), MobileNetV3-Large (0.0967), and ShuffleNet (0.1271) achieve the lowest validation loss, indicating strong generalization and effective learning. RegNetY (0.1237) and GoogleNet (InceptionV3) (0.2473) also maintain relatively low loss values, demonstrating stable validation performance. In contrast, EfficientNet-B4 (0.6626) and ResNetV2-50 (0.7047) show higher validation loss, suggesting that deeper architectures may struggle with generalization under the given dataset and training conditions. MnasNet (0.4666) and Xception (0.3278) exhibit moderate performance, while SqueezeNet (1.2855) records the highest loss, reflecting its limited capacity for complex feature learning due to extreme parameter compression. These results confirm that architectures with optimized depth and parameter efficiency, such as EfficientNet-B0 and MobileNetV3-Large, achieve the best balance between low training loss and strong generalization.

The evaluation of ten different deep learning models for macrofungi classification revealed significant variations in their classification performance, generalization capabilities, and learning efficiency. The analysis, based on key performance metrics (Table 2) and graphical representations (Figure 3, Figure 4, Figure 5, Figure 6, Figure 7 and Figure 8), provides a comprehensive understanding of how different architectures adapt to this classification task. The results indicate that lighter, well-optimized architectures generally performed better, while deeper models showed greater potential but required additional optimization or larger datasets to fully leverage their complexity. Among all models, EfficientNet-B0 emerged as the most effective classifier, achieving the highest accuracy (0.97) and F1-score (0.97), alongside a near-perfect AUC score (0.99) (Table 2). Its ROC curve (Figure 3) demonstrates a steep incline, confirming its strong class separability with minimal false positives. Additionally, its training and validation accuracy curves (Figure 5 and Figure 6) show rapid convergence, stabilizing at 98% and 97%, respectively, indicating a well-balanced learning process. The training and validation loss curves (Figure 6 and Figure 7) reinforce this, with EfficientNet-B0 reaching a final validation loss of 0.0890, one of the lowest among all models, reflecting its strong generalization capability. MobileNetV3-Large closely follows EfficientNet-B0 in terms of performance, achieving a 0.96 accuracy, 0.96 F1-score, and 0.99 AUC score (Table 2). Its ROC curve (Figure 3) demonstrates that it effectively distinguishes between macrofungi species, rivaling EfficientNet-B0. Furthermore, its training and validation accuracies (99% and 96%, respectively, in (Figure 5 and Figure 6) suggest rapid and stable convergence, making it a strong candidate for real-time applications. The training and validation loss curves (Figure 7 and Figure 8) further validate this, showing minimal loss (0.0164 and 0.0967, respectively), indicating a highly efficient learning process with strong generalization. ShuffleNet also performed remarkably well, achieving 0.95 accuracy and a 0.95 F1-score, with an AUC score of 0.97 (Table 2). Despite being a lightweight model, its ROC curve (Figure 4) follows closely behind the best-performing models, confirming its strong classification ability. The training and validation accuracy curves (Figure 5 and Figure 6) indicate fast convergence to 99% and 96%, respectively, reinforcing its efficiency as a low-complexity yet high-performing model. Furthermore, its final validation loss (0.1271, Figure 8) is among the lowest, suggesting that ShuffleNet generalizes well without overfitting. RegNetY-400MF also demonstrated competitive performance, with a 0.95 accuracy, 0.95 F1-score, and 0.98 AUC score (Table 2). The ROC curve (Figure 4) confirms its ability to separate macrofungi species with high precision. Although its training and validation accuracy (99% and 95%) suggest a slightly higher gap between training and validation, it remains one of the top-performing models. Its final validation loss (0.1237, Figure 8) indicates that while it is efficient, it may still require additional fine-tuning to enhance generalization further. GoogleNet (InceptionV3) performed slightly below the top-tier models, with a 0.93 accuracy, 0.93 F1-score, and 0.96 AUC score (Table 2). The ROC curve (Figure 3) demonstrates strong class separability but with slightly more false positives than EfficientNet-B0 or MobileNetV3-Large. While its training accuracy (99%, Figure 5) suggests rapid convergence, its validation accuracy stabilizes at 94% (Figure 6), slightly lower than that of the top models. Its training and validation loss (0.0034 and 0.2473, Figure 7 and Figure 8) indicate effective learning, though its generalization could be improved. ResNetV2-50 achieved an accuracy of 0.91, with a relatively high F1-score (0.91) and an AUC score of 0.95 (Table 2). While its ROC curve (Figure 4) suggests a solid classification ability, it did not outperform lighter models. Its training and validation accuracy curves (Figure 5 and Figure 6) show some fluctuations, with validation accuracy stabilizing at 90%, indicating possible overfitting or limited optimization. Furthermore, its training loss (0.3777, Figure 6) and validation loss (0.7047, Figure 7) are higher than those of the top-performing models, confirming that ResNetV2-50 may require additional fine-tuning or a larger dataset to improve its generalization performance. Xception demonstrated moderate performance, with a 0.88 accuracy, 0.88 F1-score, and 0.94 AUC score (Table 2). While it benefits from depthwise separable convolutions, its ROC curve (Figure 3) indicates a slightly weaker class separation compared to more optimized architectures. Its training and validation accuracy curves (Figure 5 and Figure 6) suggest relatively slow convergence, stabilizing at 93% and 88%, respectively. The final validation loss (0.3278, Figure 8) confirms that Xception requires more optimization to compete with top models. EfficientNet-B4, despite being a deeper variant of EfficientNet-B0, underperformed in this classification task. It achieved a 0.89 accuracy, 0.89 F1-score, and 0.93 AUC score (Table 2). The ROC curve (Figure 3) shows that its classification ability is weaker than its smaller counterpart, EfficientNet-B0. Its training and validation accuracy curves (Figure 5 and Figure 6) indicate slower convergence, stabilizing at 90% and 85%, respectively, suggesting that the model did not fully leverage its additional layers. Additionally, its final validation loss (0.6626, Figure 8) is significantly higher than that of EfficientNet-B0, reinforcing the idea that larger models require more data and training epochs to outperform smaller, well-optimized networks. MnasNet struggled to match the performance of the top models, achieving a 0.85 accuracy and 0.85 F1-score (Table 2). Its ROC curve (Figure 3) suggests weaker class separability, with more false positives. While its training accuracy reaches 99% (Figure 5), its validation accuracy stabilizes at only 80% (Figure 6), indicating possible overfitting and weaker generalization. Its final validation loss (0.4666, Figure 8) is relatively high, further confirming that MnasNet, while computationally efficient, does not extract features as effectively as more advanced architectures. SqueezeNet exhibited the weakest performance, with a 0.87 accuracy, 0.87 F1-score, and 0.93 AUC score (Table 2). Its ROC curve (Figure 4) highlights its higher false positive rate compared to all other models. While its training accuracy reaches 80% (Figure 5), its validation accuracy remains at 79% (Figure 6), demonstrating limited generalization. The training and validation loss curves (Figure 7 and Figure 8) reveal significantly higher loss values (1.3589 and 1.2855, respectively), confirming that SqueezeNet struggles with complex macrofungi classification due to its aggressive compression and limited feature extraction capacity. The results confirm that EfficientNet-B0, MobileNetV3-Large, ShuffleNet, and RegNetY-400MF are the most effective models for macrofungi classification, achieving the highest accuracy, strong generalization, and minimal overfitting. Deeper models like EfficientNet-B4 and ResNetV2-50 underperformed due to limited dataset size and optimization constraints, while lighter models like SqueezeNet and MnasNet struggled with feature extraction, leading to weaker generalization. The findings suggest that architectures with well-balanced depth, parameter efficiency, and optimized scaling—such as EfficientNet-B0 and MobileNetV3-Large—offer the best trade-off between accuracy, computational efficiency, and generalization in macrofungi classification tasks.

In deep learning-based classification tasks, explainable AI (XAI) techniques play a crucial role in interpreting model decisions and ensuring transparency. Among these methods, Grad-CAM (gradient-weighted class activation mapping) and Score-CAM (score-weighted class activation mapping) are widely used for visualizing which regions of an image contribute most to a model’s prediction. Grad-CAM generates heatmaps by utilizing gradient information from the final convolutional layers, highlighting the most influential image regions. However, since it relies on backpropagated gradients, it can sometimes introduce noise and less precise visualizations. Score-CAM, on the other hand, improves upon this by directly assigning importance scores to activation maps based on the model’s output, producing sharper and more reliable heatmaps without depending on gradients. These techniques help verify whether the model focuses on biologically relevant fungal structures such as texture, cap shape, and spore distribution, enhancing trust in the classification process. The following sections present Grad-CAM and Score-CAM visualizations, providing insight into how different models interpret macrofungi images.

In the heatmaps, red regions indicate the most influential areas in the model’s decision, while blue regions show the least influential ones. These visualizations demonstrate which morphological features the model focused on during classification. The Grad-CAM (Figure 9, Figure 10, Figure 11 and Figure 12) and Score-CAM (Figure 12, Figure 13, Figure 14 and Figure 15) visualizations provide a deeper understanding of how different models focus on macrofungi features during classification. These heatmaps reveal which regions contributed most to a model’s decision, with red and yellow areas indicating high relevance and blue areas showing less importance. A direct comparison between Grad-CAM and Score-CAM highlights a key observation: Score-CAM consistently produces sharper and more accurate activations, focusing on the fungal structures with minimal background interference. This is due to Score-CAM’s score-based feature map selection, which avoids gradient dependency, reducing noise and improving spatial resolution.

Among the models analyzed, ShuffleNet, Xception, and SqueezeNet demonstrate outstanding feature localization, where the red-highlighted areas precisely align with the fungal caps and stems, while the background remains largely blue. This indicates that these models effectively isolate the fungal structures, ensuring their classifications are based on relevant morphological features rather than extraneous elements. This strong performance is particularly visible in Score-CAM results (Figure 12, Figure 13, Figure 14 and Figure 15), where these models maintain concentrated focus on fungal bodies across all test images. The ability of these models to produce highly localized activations suggests that their architectural optimizations, such as depthwise separable convolutions (Xception), lightweight grouped convolutions (ShuffleNet), and parameter-efficient Fire modules (SqueezeNet), contribute to their superior spatial awareness.

On the other hand, EfficientNet-B0 and EfficientNet-B4 struggle significantly in both Grad-CAM and Score-CAM visualizations, despite their high classification accuracy. Their heatmaps exhibit wider, less focused activations, often extending into background areas or highlighting non-distinctive regions. This is particularly evident in Grad-CAM outputs (Figure 9, Figure 10, Figure 11 and Figure 12), where the attention maps appear dispersed, potentially relying on contextual clues rather than direct fungal morphology. While EfficientNet models excelled in overall classification performance, their less precise activation maps suggest that their feature extraction process may be overly reliant on contextual background cues rather than core fungal structures. This could be attributed to their compound scaling approach, which expands depth, width, and resolution uniformly, potentially diluting fine-grained feature detection in smaller, texture-dependent objects like fungi. Score-CAM results (Figure 12, Figure 13, Figure 14 and Figure 15) offer slight improvements, but EfficientNet-B0 and B4 still fail to achieve the precise, well-defined focus achieved by ShuffleNet, Xception, and SqueezeNet.

Overall, these visualizations emphasize the importance of explainable AI techniques in evaluating model reliability beyond accuracy metrics. While EfficientNet models achieve high numerical performance, their inconsistent attention maps raise concerns about their true feature learning ability. In contrast, ShuffleNet, Xception, and SqueezeNet demonstrate a strong correlation between their classification results and biologically relevant feature extraction, making them more interpretable and trustworthy for macrofungi classification tasks.

## 4. Discussion

The Discomycetes class includes fungi that play critical roles in ecosystem cycles, contributing to nutrient cycling, symbiotic relationships, and biodiversity conservation [48]. For example, species such as Geopora sumneriana and *Humaria hemisphaerica* form mycorrhizal associations in forest ecosystems, enhancing nutrient uptake and soil fertility [49]. Similarly, species like *Aleuria aurantia* and *Peziza ammophila* decompose decaying organic matter, contributing to nutrient recycling. The AI-based approaches used in this study provide valuable tools for accurately classifying these ecologically significant fungi, supporting biodiversity conservation and ecosystem management efforts [50]. Accurate classification is crucial for understanding the ecological functions of these species and their roles in maintaining ecosystem stability [51].

This study presents a novel approach to the classification of Discomycetes species using artificial intelligence (AI) and explainable artificial intelligence (XAI), moving beyond traditional methods. To enhance interpretability, the heatmaps are presented alongside the original image of each classified species, with corresponding activation maps organized by model row. This layout facilitates comparative evaluation of model attention and reliability. Fungal images were analyzed and classified using convolutional neural networks (CNNs) and Kohonen self-organizing maps (SOMs). These methods are particularly valuable for reducing data complexity while achieving high classification accuracy. Additionally, the integration of Kolmogorov–Arnold network (KAN) layers represents a significant step toward enhancing species differentiation This unique approach not only improves classification accuracy but also optimizes data processing, contributing to a better understanding of biological diversity. These findings offer digital support for traditional mycological identification processes by providing high classification accuracy for morphologically similar species. Given the morphological diversity within the Discomycetes class, the deep learning models facilitate faster and more reproducible species identification, particularly when microscopic details are critical for differentiation. For example, in this study, the EfficientNet-B0 model demonstrated the highest performance, achieving 97% accuracy, a 97% F1-score, and a 99% AUC. The MobileNetV3-L model closely followed with 96% accuracy, a 96% F1-score, and a 99% AUC, while ShuffleNet achieved 95% accuracy and a 95% F1-score. In contrast, the EfficientNet-B4 model showed relatively lower performance with 89% accuracy, an 89% F1-score, and a 93% AUC (Table 2). These findings offer digital support to traditional mycological identification processes by providing high classification accuracy for morphologically similar species. Given the morphological diversity within the Discomycetes class, the deep learning models facilitate faster and more reproducible species identification, particularly when microscopic details are critical for differentiation.

The methods employed in this study include CNNs [52], SOMs [53], and KAN [54] layers, each offering unique advantages. CNNs effectively extract local and global features from images, while SOMs excel at clustering data with complex, nonlinear structures [55]. In contrast, KAN layers, applied for the first time in this study for fungal classification, offer advanced modeling capabilities for intricate data patterns [14]. For instance, the EfficientNet-B0 model, which achieved 97% accuracy, a 97% F1-score, and a 99% AUC, demonstrated significant advantages in biologically dense imaging studies [56]. On the other hand, lighter models like ShuffleNet, with 95% accuracy and a 95% F1-score, offer high accuracy with lower computational costs and faster processing times (Table 2) [57]. The limited dataset size may pose an overfitting risk for deep learning models. To mitigate this, extensive data augmentation and early stopping techniques were employed. Moreover, the parallel trends in training and test performance support the generalization capability of the models.

An analysis of the algorithms used in the literature reveals that CNNs are widely employed in various applications, including medical image processing, remote sensing, object recognition, and biological tissue analysis [58]. CNN-based deep learning models, in particular, have demonstrated high accuracy rates in complex classification tasks, especially in medical diagnostics and bioinformatics [59]. In this study, the EfficientNet-B0 model achieved 97% accuracy, a 97% F1-score, and a 99% AUC, outperforming similar studies in the literature [60]. The MobileNetV3-L model also produced impressive results, with 96% accuracy, a 96% F1-score, and a 99% AUC, highlighting the effectiveness of AI-based approaches in biological data classification (Table 2).

## 5. Conclusions

In this study, the performance of deep learning models used for the classification of Discomycetes species was comprehensively evaluated. EfficientNet-B0 emerged as the best-performing model, achieving 97% accuracy, a 97% F1-score, and a 99% AUC, highlighting its critical importance for accurate biodiversity classification and conservation. The MobileNetV3-L model followed closely with 96% accuracy, a 96% F1-score, and a 99% AUC, while ShuffleNet also demonstrated impressive results with 95% accuracy and a 95% F1-score. In contrast, the EfficientNet-B4 model showed lower performance, with 89% accuracy, an 89% F1-score, and a 93% AUC, indicating its relative disadvantage in extracting fine-grained biological features. These results underscore the superior feature extraction capabilities and overall accuracy of EfficientNet-B0 and MobileNetV3-L.

Explainable artificial intelligence (XAI) techniques, particularly Grad-CAM and Score-CAM, played a crucial role in this study by providing transparency into the decision-making processes of deep learning models. These methods significantly enhanced the interpretability of classification decisions, ensuring the reliability of AI systems in biological data analysis. The high performance of the EfficientNet-B0 model, combined with Score-CAM-based visualizations, demonstrates its robustness and reliability for biological classification tasks.

For future research, the development of more precise classification systems using larger datasets and more advanced AI models is recommended in fields such as biological data analysis, ecosystem modeling, bioinformatics, agricultural technology, environmental monitoring, and medical imaging. Specifically, applications in microorganism species identification, disease diagnosis, genetic data analysis, and biodiversity monitoring stand to benefit significantly from explainable AI approaches. The findings of this study can guide the development of these systems, setting new standards for biological data analysis.

## Figures and Tables

**Figure 1 biology-14-00719-f001:**
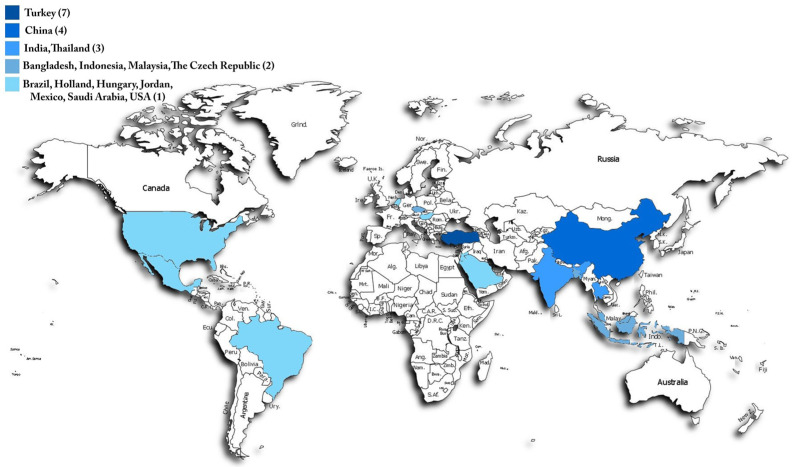
Countries conducting AI-supported macrofungus classification studies and the number of published articles.

**Figure 2 biology-14-00719-f002:**
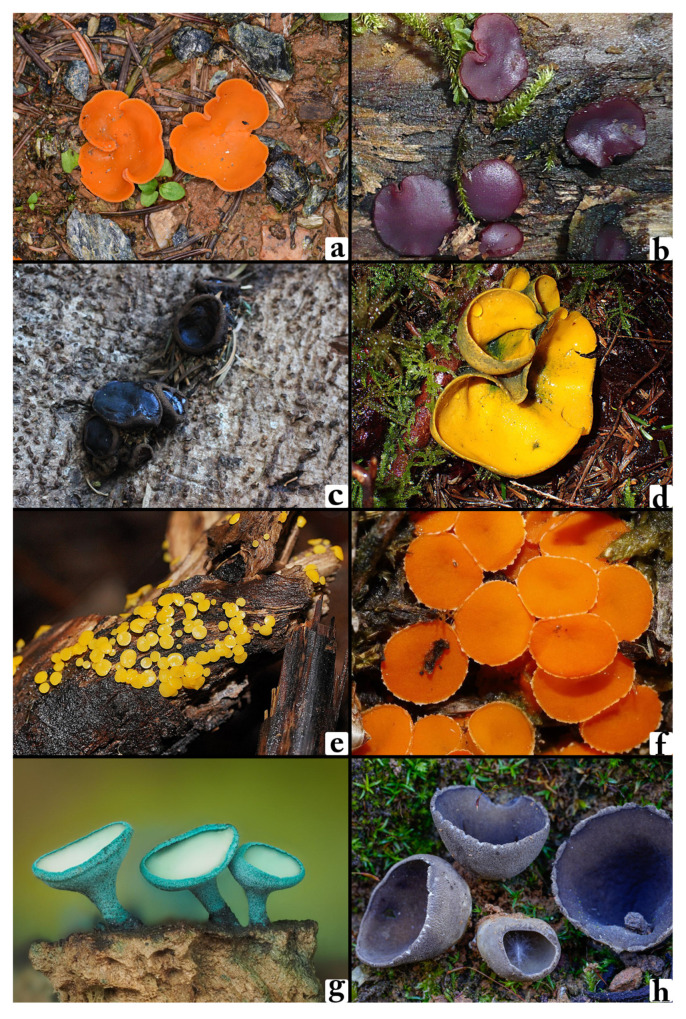
The dataset employed in this study includes 14 different macrofungal species. (**a**) *Aleuria aurantia*, (**b**) *Ascocoryne cylichnium*, (**c**) *Bulgaria inquinans*, (**d**) *Caloscypha fulgens*, (**e**) *Calycina citrina*, (**f**) *Cheilymenia granulata*, (**g**) *Chlorociboria aeruginosa*, (**h**) *Dissingia leucomelaena*, (**i**) *Geophora sumneriana*, (**j**) *Humaria hemisphaerica*, (**k**) *Lanzia echinophila*, (**l**) *Paragalactinia succosa,* (**m**) *Peziza ammophila*, and (**n**) *Sarcosphaera coronaria*.

**Figure 3 biology-14-00719-f003:**
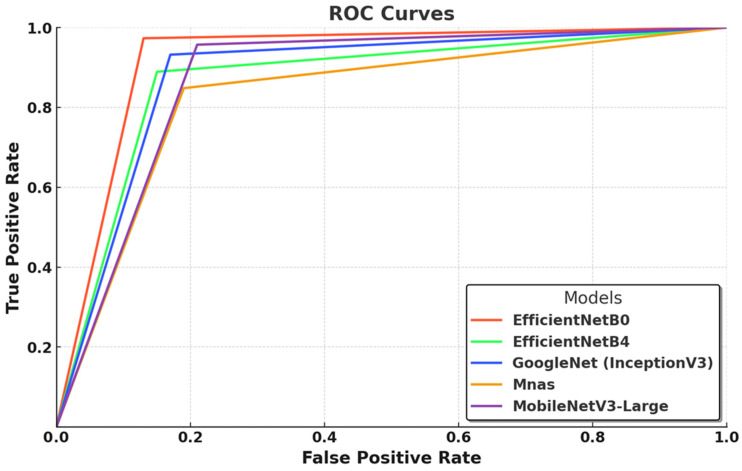
ROC curves of EfficientNetB0, EfficientNetB4, GoogleNet, Mnas, and MobileNetV3-Large.

**Figure 4 biology-14-00719-f004:**
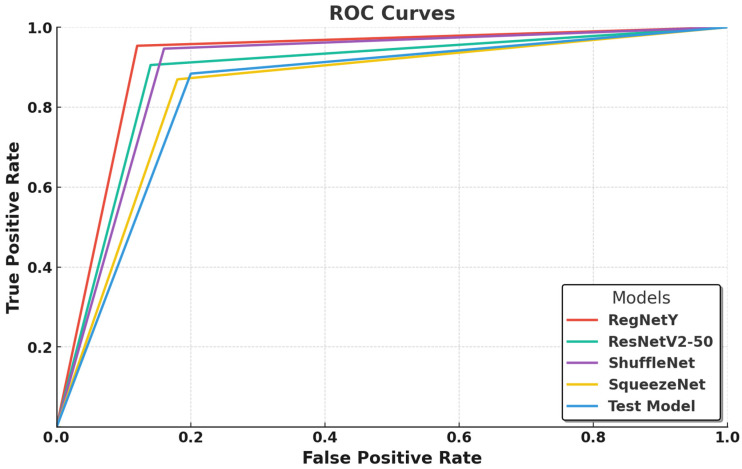
ROC curves of EfficientNetB0, EfficientNetB4, GoogleNet, Mnas, and MobileNetV3-Large.

**Figure 5 biology-14-00719-f005:**
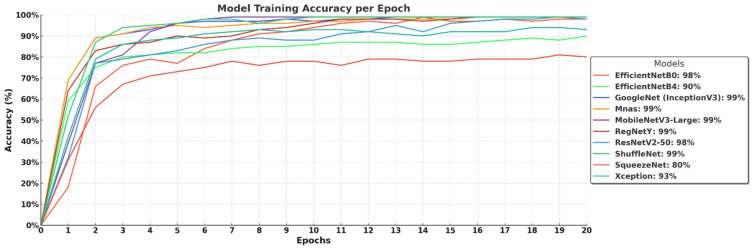
Training accuracy of deep learning models over 20 epochs.

**Figure 6 biology-14-00719-f006:**
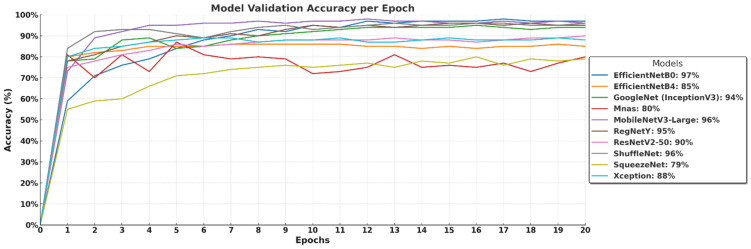
Validation accuracy of deep learning models over 20 epochs.

**Figure 7 biology-14-00719-f007:**
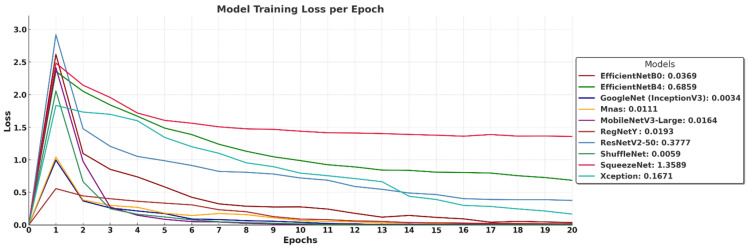
Training loss of deep learning models over 20 epochs.

**Figure 8 biology-14-00719-f008:**
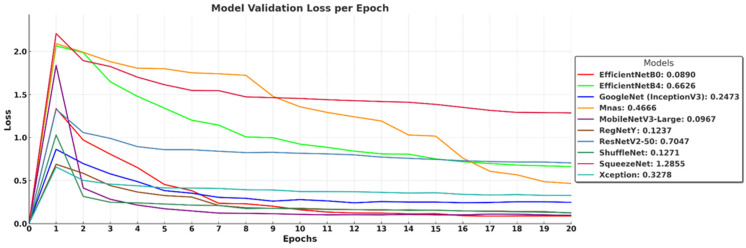
Validation loss of deep learning models over 20 epochs.

**Figure 9 biology-14-00719-f009:**
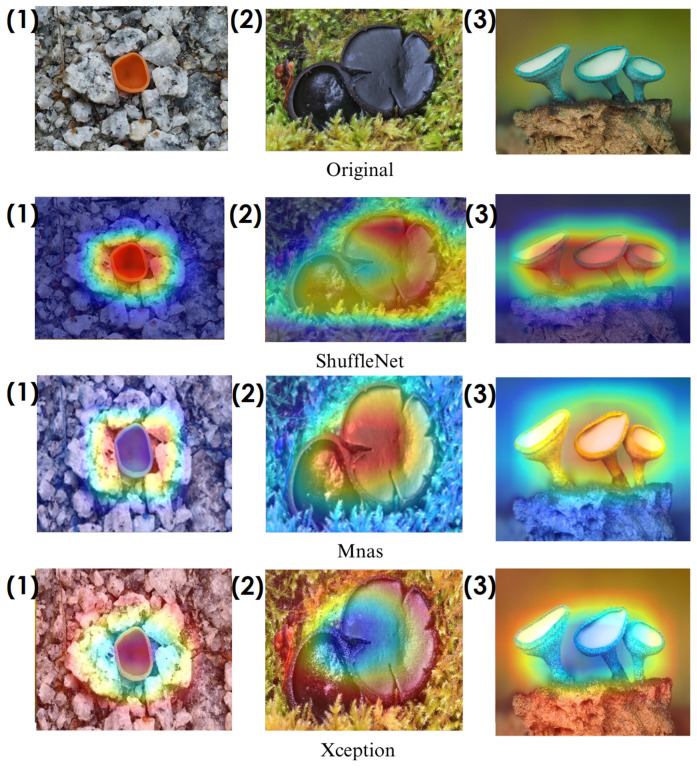
Grad-CAM visualizations for macrofungi classification using ShuffleNet, Mnas, and Xception models. The top row presents the original images of (1) *Aleuria aurantia*, (2) *Bulgaria inquinans*, and (3) *Chlorociboria aeruginosa* from left to right. The second, third, and fourth rows display the corresponding activation maps for ShuffleNet, Mnas, and Xception, respectively. These visualizations highlight the most influential regions used by the models for classification. Red areas in the heatmaps indicate the most influential regions for classification, while blue areas indicate low influence. The layout presents the original image followed by model-specific activation maps, improving interpretability.

**Figure 10 biology-14-00719-f010:**
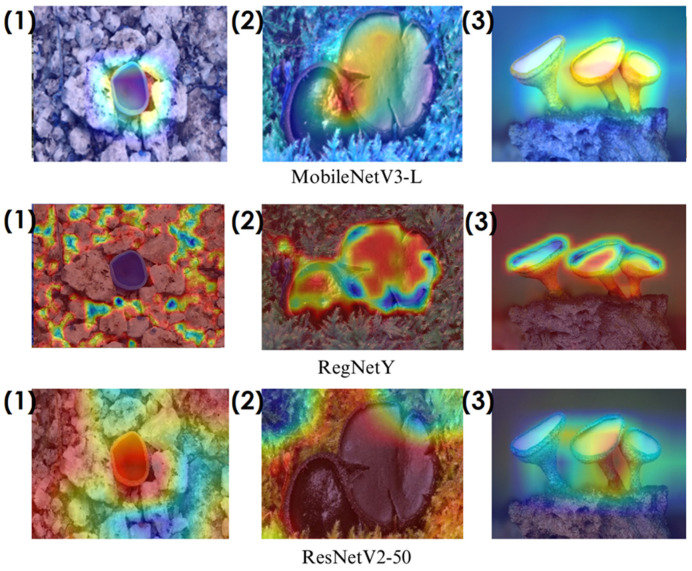
Grad-CAM visualizations for macrofungi classification using MobileNetV3-L, RegNetY, and ResNetV2-50 models. The images correspond to three different macrofungi species: (1) *Aleuria aurantia*, (2) *Bulgaria inquinans*, and (3) *Chlorociboria aeruginosa* from left to right. The rows display the corresponding activation maps for MobileNetV3-L, RegNetY, and ResNetV2-50, respectively. These visualizations highlight the most influential regions used by the models for classification.

**Figure 11 biology-14-00719-f011:**
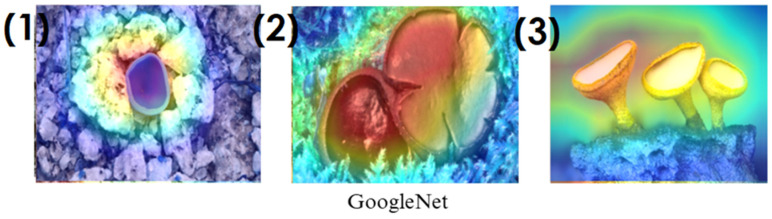
Grad-CAM visualizations for macrofungi classification using the GoogleNet model. The images correspond to three different macrofungi species: (1) *Aleuria aurantia*, (2) *Bulgaria inquinans*, and (3) *Chlorociboria aeruginosa* from left to right. The row displays the corresponding activation maps for GoogleNet. These visualizations highlight the most influential regions used by the model for classification. Red areas in the heatmaps indicate the most influential regions for classification, while blue areas indicate low influence. The layout presents the original image followed by model-specific activation maps, improving interpretability.

**Figure 12 biology-14-00719-f012:**
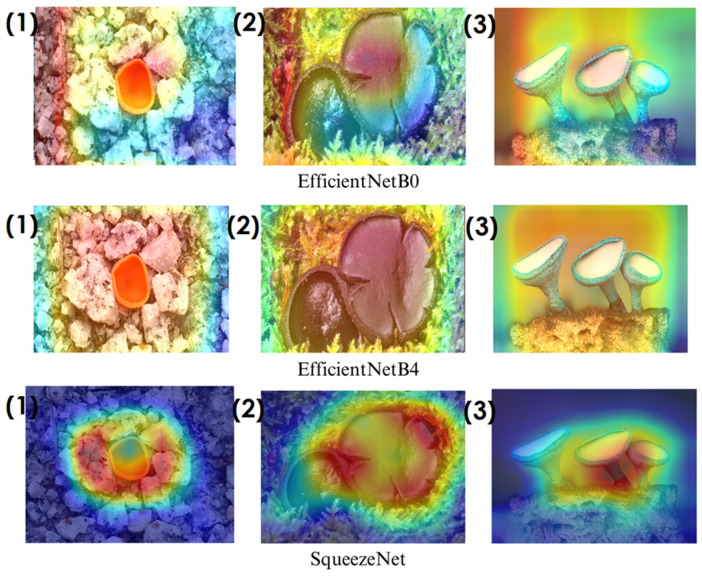
Grad-CAM visualizations for macrofungi classification using EfficientNetB0, EfficientNetB4, and SqueezeNet models. The images correspond to three different macrofungi species: (1) *Aleuria aurantia*, (2) *Bulgaria inquinans*, and (3) *Chlorociboria aeruginosa* from left to right. The rows display the corresponding activation maps for EfficientNetB0, EfficientNetB4, and SqueezeNet, respectively. These visualizations highlight the most influential regions used by the models for classification. Red areas in the heatmaps indicate the most influential regions for classification, while blue areas indicate low influence. The layout presents the original image followed by model-specific activation maps, improving interpretability.

**Figure 13 biology-14-00719-f013:**
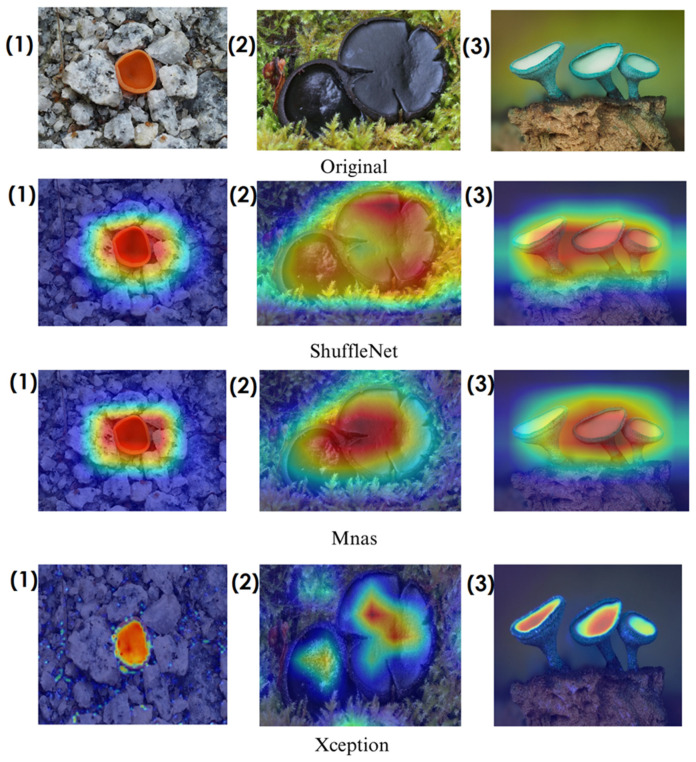
Score-CAM visualizations for macrofungi classification using ShuffleNet, Mnas, and Xception models. The top row presents the original images of (1) *Aleuria aurantia*, (2) *Bulgaria inquinans*, and (3) *Chlorociboria aeruginosa* from left to right. The second, third, and fourth rows display the corresponding activation maps for ShuffleNet, Mnas, and Xception, respectively. These visualizations highlight the most influential regions used by the models for classification. Red areas in the heatmaps indicate the most influential regions for classification, while blue areas indicate low influence. The layout presents the original image followed by model-specific activation maps, improving interpretability.

**Figure 14 biology-14-00719-f014:**
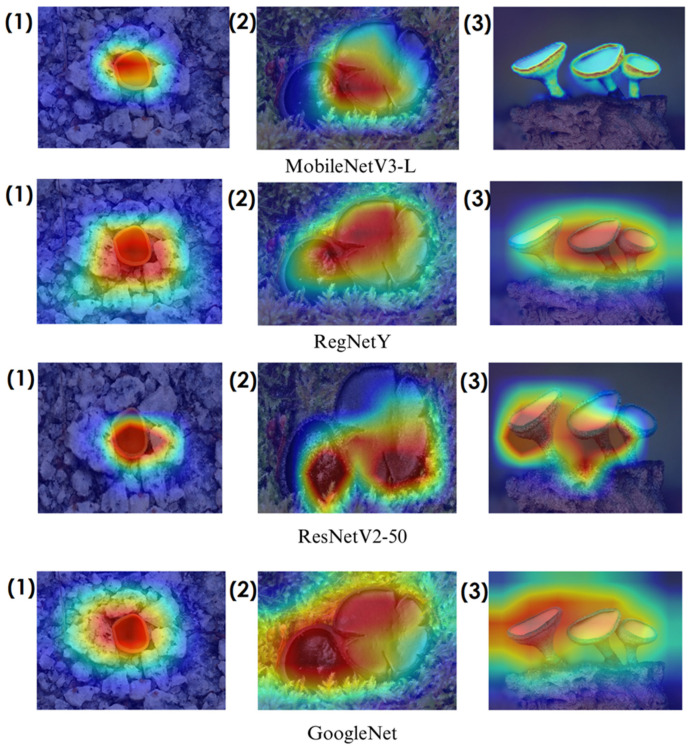
Score-CAM visualizations for macrofungi classification using MobileNetV3-L, RegNetY, ResNetV2-50, and GoogleNet models. The images correspond to three different macrofungi species: (1) *Aleuria aurantia*, (2) *Bulgaria inquinans*, and (3) *Chlorociboria aeruginosa* from left to right. The rows display the corresponding activation maps for MobileNetV3-L, RegNetY, ResNetV2-50, and GoogleNet, respectively. These visualizations highlight the most influential regions used by the models for classification. Red areas in the heatmaps indicate the most influential regions for classification, while blue areas indicate low influence. The layout presents the original image followed by model-specific activation maps, improving interpretability.

**Figure 15 biology-14-00719-f015:**
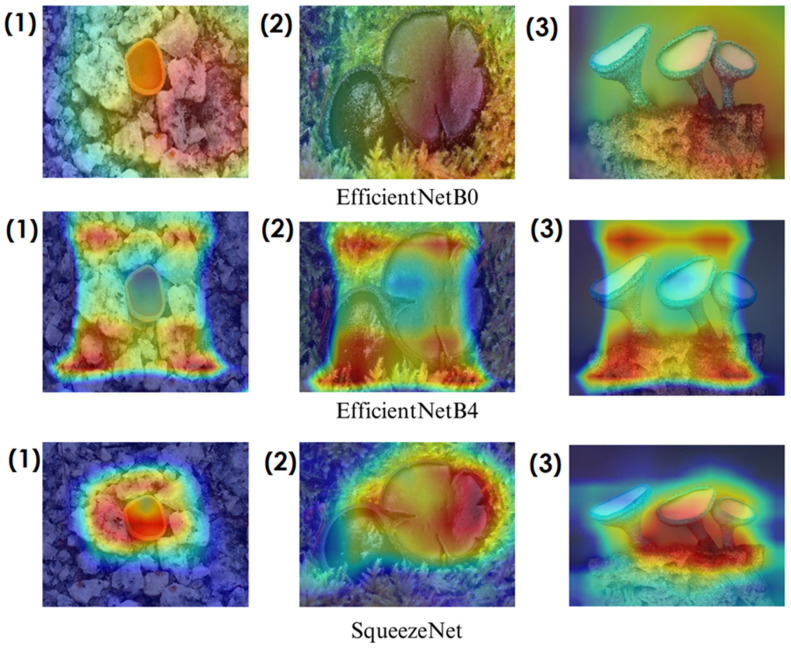
Score-CAM visualizations for macrofungi classification using EfficientNetB0, EfficientNetB4, and SqueezeNet models. The images correspond to three different macrofungi species: (1) *Aleuria aurantia*, (2) *Bulgaria inquinans*, and (3) *Chlorociboria aeruginosa* from left to right. The rows display the corresponding activation maps for EfficientNetB0, EfficientNetB4, and SqueezeNet, respectively. These visualizations highlight the most influential regions used by the models for classification. Red areas in the heatmaps indicate the most influential regions for classification, while blue areas indicate low influence. The layout presents the original image followed by model-specific activation maps, improving interpretability.

**Table 1 biology-14-00719-t001:** Names of datasets obtained from open sources, their respective sources, the proportion each dataset contributes to the overall dataset, and the technical details of these sources [15].

Mushroom Species Name	% of Photographs Received from Source	Types of Photographs	Resoulution	Source URL
*Aleuria aurantia*	<78	JPEG	300 dpi	www.gbif.org
*Ascocoryne cylichnium*	<80	JPEG	300 dpi	www.gbif.org
*Bulgaria inquinans*	<77	JPEG	300 dpi	www.gbif.org
*Caloscypha fulgens*	<82	JPEG	300 dpi	www.gbif.org
*Calycina citrina*	<81	JPEG	300 dpi	www.gbif.org
*Cheilymenia granulata*	<80	JPEG	300 dpi	www.gbif.org
*Chlorociboria aeruginosa*	<78	JPEG	300 dpi	www.gbif.org
*Dissingia leucomelaena*	<78	JPEG	300 dpi	www.gbif.org
*Geophora sumneraiana*	<82	JPEG	300 dpi	www.gbif.org
*Humaria hemisphaerica*	<81	JPEG	300 dpi	www.gbif.org
*Lanzia echinophila*	<81	JPEG	300 dpi	www.gbif.org
*Paragalactinia succosa*	<80	JPEG	300 dpi	www.gbif.org
*Peziza ammophila*	<79	JPEG	300 dpi	www.gbif.org
*Sarcosphaera coronaria*	<77	JPEG	300 dpi	www.gbif.org

**Table 2 biology-14-00719-t002:** Performance comparison of deep learning models based on accuracy, precision, recall, F1-score, specificity, and AUC score.

Models	Accuracy	Precision	Recall	F1-Score	Specificity	AUC
**ShuffleNet**	**0.95**	**0.95**	**0.95**	**0.95**	**0.99**	**0.97**
MnasNet	0.85	0.87	0.85	0.85	0.99	0.92
GoogleNet	0.93	0.94	0.93	0.93	0.99	0.96
Xception	0.88	0.89	0.88	0.88	0.99	0.94
**MobileNetV3-L**	**0.96**	**0.96**	**0.96**	**0.96**	**0.99**	**0.99**
**RegNetY**	**0.95**	**0.95**	**0.95**	**0.95**	**0.99**	**0.98**
ResNetV2-50	0.91	0.91	0.91	0.91	0.99	0.95
**EfficientNetB0**	**0.97**	**0.97**	**0.97**	**0.97**	**0.99**	**0.99**
EfficientNet-B4	0.89	0.89	0.89	0.89	0.99	0.93
SqueezeNet	0.87	0.87	0.87	0.87	0.99	0.93

## Data Availability

The raw data supporting the conclusions of this article will be made available by the authors on request.

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
