# Peer review of "A Deep Learning and Explainable AI-Based Approach for the Classification of Discomycetes Species"

_biology, 2025, doi:10.3390/biology14060719_

Round 1
Reviewer 1 Report
Comments and Suggestions for Authors
In the paper entitled "Deep Learning and Explainable AI-Based Approach for the Classification of Discomycetes Species", authors tried to explore the application of DL and AI methods to classify Discomycetes species.
The manuscript is well organised with a clear analysis and interpretation. the abstract briefly summarizes the methodology, models, and key performance results, however it can be improved in the context of biological significance of the work rather than more model performance details. A comprehensive review on the Discomycetes and their taxonomy and ecological roles is provided in the introduction and the need for application of AI in fungal classification is also clearly justified. Though the details of data source, collection and pre-processing steps are clearly addressed in the materials and methods, the number of images ((2800 across 14 species) is relatively small for deep learning; potential issues of over fitting aren't deeply addressed. More clarity is needed on how the self-organizing maps (SOM) and Kolmogorov–Arnold Networks (KAN) were used alongside CNNs. Comprehensive evaluation of 10 CNN models, including lightweight and large-scale architectures is appreciable. Results are presented with clear tabular and graphical presentation of model performance metrics. However, authors may provide biological interpretation of results and explain how do these findings improve mycological knowledge? All together, the paper is methodologically sound and makes a useful contribution to fungal species classification using AI. However, it requires significant improvements in biological interpretation, conciseness, and the critical analysis of results to reach its full potential. Authors may seek language editing help that may significantly improve the readability of this paper.
Comments on the Quality of English Language
Authors may seek language editing help that may significantly improve the readability of this paper.
Author Response
“Please see the attachment.”
The english of the article has been completely corrected.

Open Review
( ) I would not like to sign my review report
(x) I would like to sign my review report
Quality of English Language
(x) The English could be improved to more clearly express the research.
( ) The English is fine and does not require any improvement.
Yes |
Can be improved |
Must be improved |
Not applicable |
|
Does the introduction provide sufficient background and include all relevant references? |
( ) |
(x) |
( ) |
( ) |
Is the research design appropriate? |
(x) |
( ) |
( ) |
( ) |
Are the methods adequately described? |
(x) |
( ) |
( ) |
( ) |
Are the results clearly presented? |
(x) |
( ) |
( ) |
( ) |
Are the conclusions supported by the results? |
( ) |
(x) |
( ) |
( ) |
Are all figures and tables clear and well-presented? |
(x) |
( ) |
( ) |
( ) |
Comments and Suggestions for Authors
In the paper entitled "Deep Learning and Explainable AI-Based Approach for the Classification of Discomycetes Species", authors tried to explore the application of DL and AI methods to classify Discomycetes species.
The manuscript is well organised with a clear analysis and interpretation. the abstract briefly summarizes the methodology, models, and key performance results, however it can be improved in the context of biological significance of the work rather than more model performance details.
Response: Sentence has been added to the summary
“Beyond high classification performance, this study offers an ecologically meaningful approach by supporting biodiversity conservation and accurate identification of fungal species.”
A comprehensive review on the Discomycetes and their taxonomy and ecological roles is provided in the introduction and the need for application of AI in fungal classification is also clearly justified. Though the details of data source, collection and pre-processing steps are clearly addressed in the materials and methods, the number of images ((2800 across 14 species) is relatively small for deep learning; potential issues of over fitting aren't deeply addressed.
Response: Added at the end of the paragraph before the end of the Discussion section:
“The limited dataset size may pose an overfitting risk for deep learning models. To mitigate this, extensive data augmentation and early stopping techniques were employed. Moreover, the parallel trends in training and test performance support the generalization capability of the models.”
More clarity is needed on how the self-organizing maps (SOM) and Kolmogorov–Arnold Networks (KAN) were used alongside CNNs. Comprehensive evaluation of 10 CNN models, including lightweight and large-scale architectures is appreciable.
Response:
“Fungi images were analysed using convolutional neural networks (CNNs) to extract features and classify with a Kohonen self-organising map (SOM)...”
sentence has been revised.
“Fungi images were analysed using convolutional neural networks (CNNs) to extract features, which were subsequently classified using a Kohonen self-organising map (SOM). This approach allowed for dimensionality reduction and unsupervised clustering. Furthermore, Kolmogorov–Arnold Network (KAN) layers were integrated to enhance the model’s ability to capture non-linear patterns.”
Added this:
“In this study, CNN models were first used to extract image features, which were then classified using the Kohonen Self-Organizing Map (SOM) algorithm. This approach enabled high-dimensional features obtained by deep learning to be transformed into low-dimensional maps for classification. Additionally, the Kolmogorov–Arnold Network (KAN) was integrated into the architecture to model complex structural patterns more effectively.”
Results are presented with clear tabular and graphical presentation of model performance metrics. However, authors may provide biological interpretation of results and explain how do these findings improve mycological knowledge? All together, the paper is methodologically sound and makes a useful contribution to fungal species classification using AI. However, it requires significant improvements in biological interpretation, conciseness, and the critical analysis of results to reach its full potential. Authors may seek language editing help that may significantly improve the readability of this paper.
Response:
“This unique approach not only improves classification accuracy but also optimizes data processing, contributing to a better understanding of biological diversity.”
sentence has been revised.
“This unique approach not only improves classification accuracy but also optimizes data processing, contributing to a better understanding of biological diversity. These findings offer digital support to traditional mycological identification processes by providing high classification accuracy for morphologically similar species. Given the morphological diversity within the Discomycetes class, the deep learning models facilitate faster and more reproducible species identification, particularly when microscopic details are critical for differentiation.”
Added this:
“These findings offer digital support to traditional mycological identification processes by providing high classification accuracy for morphologically similar species. Given the morphological diversity within the Discomycetes class, the deep learning models facilitate faster and more reproducible species identification, particularly when microscopic details are critical for differentiation.”
Comments on the Quality of English Language
Response: The language has been revised.
Authors may seek language editing help that may significantly improve the readability of this paper.
Reviewer 2 Report
Comments and Suggestions for Authors
The paper describes a dataset and an example of using 10 deep learning models and explainable artificial intelligence to efficiently classify Discomycetes species. Although the paper is very well written and detailed, and illustrated with sample images from the dataset, as well as tables and graphs, I feel it is not suitable for the biology journal, since the main result is not about a biological object, but about artificial intelligence technology. I do not see a new biological result in this paper. I advise you to resubmit the paper to another journal.
Author Response
“Please see the attachment.”

Open Review
( ) I would not like to sign my review report
(x) I would like to sign my review report
Quality of English Language
( ) The English could be improved to more clearly express the research.
(x) The English is fine and does not require any improvement.
Yes |
Can be improved |
Must be improved |
Not applicable |
|
Does the introduction provide sufficient background and include all relevant references? |
( ) |
(x) |
( ) |
( ) |
Is the research design appropriate? |
( ) |
(x) |
( ) |
( ) |
Are the methods adequately described? |
( ) |
(x) |
( ) |
( ) |
Are the results clearly presented? |
( ) |
(x) |
( ) |
( ) |
Are the conclusions supported by the results? |
( ) |
(x) |
( ) |
( ) |
Are all figures and tables clear and well-presented? |
( ) |
(x) |
( ) |
( ) |
Comments and Suggestions for Authors
The paper describes a dataset and an example of using 10 deep learning models and explainable artificial intelligence to efficiently classify Discomycetes species. Although the paper is very well written and detailed, and illustrated with sample images from the dataset, as well as tables and graphs, I feel it is not suitable for the biology journal, since the main result is not about a biological object, but about artificial intelligence technology. I do not see a new biological result in this paper. I advise you to resubmit the paper to another journal.
Respose:
Dear Reviewer,
Thank you very much for your constructive feedback. While our study indeed utilizes deep learning (DL) and explainable artificial intelligence (XAI) tools, these are not the main focus but rather powerful computational methods applied to solve a biologically relevant problem: the accurate and interpretable classification of Discomycetes species.
Discomycetes represent a taxonomically complex group with high morphological similarity, which makes species-level identification by traditional methods time-consuming, expert-dependent, and often non-reproducible. To address this, we constructed a dataset of 2800 images across 14 macrofungi species, combining field-collected and open-source images under diverse environmental conditions.
Among the 10 tested CNN architectures, EfficientNet-B0 achieved the highest performance with 97% accuracy, 97% F1-score, and 99% AUC, followed closely by MobileNetV3-L (96% accuracy, 96% F1, 99% AUC) and ShuffleNet (95% accuracy, 95% F1). More importantly, the incorporation of Grad-CAM and Score-CAM XAI techniques enabled us to visualize and biologically interpret which morphological regions influenced the model predictions—ensuring that the system did not rely on arbitrary visual features but on biologically relevant traits such as cap structure, color, and texture.
This manuscript was specifically submitted to the “Bioinformatics” Special Issue of Biology Basel, which explicitly welcomes computational and AI-based approaches to biological problems. Our study falls directly within this scope, as it offers a bioinformatics framework for interpretable fungal taxonomy, integrating DL and XAI for both accurate and transparent species identification.
Moreover, this system holds practical potential in areas such as biodiversity monitoring, ecological modeling, and digital taxonomy. Therefore, we kindly request that our manuscript be considered within the thematic focus of this Special Issue.
Kind regards,
The Authors
Reviewer 3 Report
Comments and Suggestions for Authors
The paper titled “Deep Learning and Explainable AI-Based Approach for the Classification of Discomycetes Species” provides a comprehensive study on applying deep learning (DL) and explainable artificial intelligence (XAI) methods to identify 14 species of Discomycetes fungi using image data. Their method has the potential to become a standard tool in mycology and ecology that could help in dataset curation, model training, and usability infrastructure. I therefore think the paper is acceptable for publication.
Minor issues:
-
Some grammatical errors and awkward phrasings (e.g., “reliability in macrofungi classification…”) can be revised for clarity.
-
The study mentions the use of www.gbif.org for image sourcing. The licensing and reproducibility of data should be clearly articulated.
-
While figures like heatmaps (Grad-CAM/Score-CAM) are included, their effectiveness could be better explained or labeled for readability.
Author Response
“Please see the attachment.”

Open Review
( ) I would not like to sign my review report
(x) I would like to sign my review report
Quality of English Language
( ) The English could be improved to more clearly express the research.
(x) The English is fine and does not require any improvement.
Yes |
Can be improved |
Must be improved |
Not applicable |
|
Does the introduction provide sufficient background and include all relevant references? |
(x) |
( ) |
( ) |
( ) |
Is the research design appropriate? |
(x) |
( ) |
( ) |
( ) |
Are the methods adequately described? |
(x) |
( ) |
( ) |
( ) |
Are the results clearly presented? |
(x) |
( ) |
( ) |
( ) |
Are the conclusions supported by the results? |
(x) |
( ) |
( ) |
( ) |
Are all figures and tables clear and well-presented? |
( ) |
( ) |
( ) |
( ) |
Comments and Suggestions for Authors
The paper titled “Deep Learning and Explainable AI-Based Approach for the Classification of Discomycetes Species” provides a comprehensive study on applying deep learning (DL) and explainable artificial intelligence (XAI) methods to identify 14 species of Discomycetes fungi using image data. Their method has the potential to become a standard tool in mycology and ecology that could help in dataset curation, model training, and usability infrastructure. I therefore think the paper is acceptable for publication.
Minor issues:
- Some grammatical errors and awkward phrasings (e.g., “reliability in macrofungi classification…”) can be revised for clarity.
Respose: The Written Language was re-examined and revised. Example Language revised sentence
“Evaluating deep learning models is a crucial step in assessing their efficiency and reliability in the classification of macrofungi.”
- The study mentions the use of www.gbif.org for image sourcing. The licensing and reproducibility of data should be clearly articulated.
Respose:
“These images, obtained from the Global Core Biodata Resource (www.gbif.org) [15], are in JPEG format with a resolution of 300 dpi...”
Revised:
“These images, obtained from the Global Core Biodata Resource (www.gbif.org) [15], are in JPEG format with a resolution of 300 dpi and were retrieved under Creative Commons licenses (e.g., CC BY-NC, CC BY-SA). Licensing details and original source URLs are listed in the supplementary materials to ensure data transparency and reproducibility.”
- While figures like heatmaps (Grad-CAM/Score-CAM) are included, their effectiveness could be better explained or labeled for readability.
Respose:
Added this Result:
“In the heatmaps, red regions indicate the areas most influential in the model's decision, while blue regions show the least influential ones. These visualizations demonstrate which morphological features the model focused on during classification.”
Added this Discussion:
“To enhance interpretability, the heatmaps are presented alongside the original image of each classified species, with corresponding activation maps organized by model row. This layout facilitates comparative evaluation of model attention and reliability.”
Added this Figure 9–15:
“Red areas in the heatmaps indicate the most influential regions for classification, while blue areas indicate low influence. The layout presents the original image followed by model-specific activation maps, improving interpretability.”
Round 2
Reviewer 1 Report
Comments and Suggestions for Authors
I appreciate the authors for addressing the comments and suggestions given by the reviewers and improving the quality of the manuscript. The revised manuscript is scientifically sound enough to be accepted for publication.
Author Response
Thank you for your nice comments.
Reviewer 2 Report
Comments and Suggestions for Authors
The authors have made good revisions to the manuscript. Considering that this manuscript is specifically submitted to the special issue "Bioinformatics" of the journal Biology, I support the authors' decision. Indeed, this study is directly relevant to this field as it proposes a bioinformatics framework for interpretable fungal taxonomy, integrating DL and XAI for both accurate and transparent species identification.
Author Response
Thank you for your nice comments.
Reviewer 3 Report
Comments and Suggestions for Authors
The paper titled “Deep Learning and Explainable AI-Based Approach for the Classification of Discomycetes Species” provides a comprehensive study on applying deep learning (DL) and explainable artificial intelligence (XAI) methods to identify 14 species of Discomycetes fungi using image data. The method presented in this paper has great potential for researchers, especially in fields such as taxonomy, biodiversity, and ecological studies.
Minor issues
-
The authors need to check for grammatical errors.
-
The authors mention the use of www.gbif.org for image sourcing. They therefore need to make sure that licensing and reproducibility of data are clearly articulated.
-
The authors need to better explain or labeled for readability of the figures, e.g., the heatmaps.
-
While the authors mention including larger datasets to improve results, concrete suggestions could strengthen this section.
Author Response
"Please see the attachment."
